# Enzymatic synthesis of mono- and trifluorinated alanine enantiomers expands the scope of fluorine biocatalysis
Manuel Nieto-Domínguez[1], Aboubakar Sako[1], Kasper Enemark-Rasmussen[2], Charlotte Held Gotfredsen[2], Daniela Rago[1] & Pablo I. Nikel [1]✉

Fluorinated amino acids serve as an entry point for establishing new-to-Nature chemistries in biological systems, and novel methods are needed for the selective synthesis of these building blocks. In this study, we focused on the enzymatic synthesis of fluorinated alanine enantiomers to expand fluorine biocatalysis. The alanine dehydrogenase from *Vibrio proteolyticus* and the diaminopimelate dehydrogenase from *Symbiobacterium thermophilum* were selected for in vitro production of (*R*)-3-fluoroalanine and (*S*)-3-fluoroalanine, respectively, using 3-fluoropyruvate as the substrate. Additionally, we discovered that an alanine racemase from *Streptomyces lavendulae*, originally selected for setting an alternative enzymatic cascade leading to the production of these non-canonical amino acids, had an unprecedented catalytic efficiency in β-elimination of fluorine from the monosubstituted fluoroalanine. The in vitro enzymatic cascade based on the dehydrogenases of *V. proteolyticus* and *S. thermophilum* included a cofactor recycling system, whereby a formate dehydrogenase from *Pseudomonas* sp. 101 (either native or engineered) coupled formate oxidation to NAD(P)H formation. Under these conditions, the reaction yields for (*R*)-3-fluoroalanine and (*S*)-3-fluoroalanine reached >85% on the fluorinated substrate and proceeded with complete enantiomeric excess. The selected dehydrogenases also catalyzed the conversion of trifluoropyruvate into trifluorinated alanine as a first-case example of fluorine biocatalysis with amino acids carrying a trifluoromethyl group.

The extensive array of chemicals essential to our contemporary society is predominantly supplied by the chemical industry. These chemicals include polymers, agrochemicals and pharmaceutical molecules, the production of which largely depends on traditional synthetic chemistry. Conventional approaches for the synthesis of these compounds often employ processes and reagents leading to significant environmental hazards, notably through the generation of toxic by-products and waste streams[1,2]. With increasing environmental awareness, there is a growing interest in leveraging Nature and biosynthesis as alternative sources for fulfilling chemical production needs[3,4]. The molecular diversity in the biosphere is considered nearly boundless, yet only a small portion of these biomolecules has been explored so far[5]. Nonetheless, the correlation between industrially-relevant chemicals and naturally-occurring products is remarkably limited[6]. A key factor contributing to this disparity is the industrial reliance on synthetic compounds containing chemical elements, e.g., fluorine (F)[7] and other halogen atoms[8], which are not typically found in natural biological systems[9,10]. Establishing a biotechnological alternative to chemical synthesis requires the rational design of biosynthetic pathways and degradation routes either as enzymatic cascades in vitro[11,12] or as part of living organisms[13], enabling them to execute new-to-Nature chemistries[14,15].

Amino acids represent an attractive target for bioproduction due to their wide industrial exploitation[16]. The twenty standard proteinogenic amino acids, while relatively simple in structure, are fundamental components of all naturally-occurring polypeptides. Furthermore, the D-enantiomers of some amino acids are essential in the formation of bacterial peptidoglycans[17] and are involved in the biosynthesis of natural peptide antibiotics[18]. In this sense, introducing the D-enantiomers of non-canonical amino acids (NCAAs) into the chemistry of living bacterial cells

[1]The Novo Nordisk Foundation Center for Biosustainability, Technical University of Denmark, Kongens Lyngby, Denmark. [2]Department of Chemistry, NMR Center, Technical University of Denmark, Kongens Lyngby, Denmark. ✉e-mail: pabnik@biosustain.dtu.dk

represents an effective approach to substantially enhance the chemical diversity of cellular structures[19–21], but the biocatalysis toolbox for their production is somewhat limited[22,23]. Among the broad group of NCAAs, fluorinated amino acids (FAAs), which contain one or more F atoms, have considerable potential for engineering new chemistries. The small size of the F atom renders FAAs structurally similar to their natural analogues, making them largely indistinguishable by the cellular machinery[24]. Currently, most FAAs are produced through chemical synthesis[25], although some of them can be obtained enzymatically in vitro[24,26]. Notably, 4-fluoro-L-threonine is a naturally-occurring FAA produced by *Streptomyces cattleya* and other actinomycetes[27]. To date, research approaches for the production of FAAs have primarily focused on the L-enantiomers, exploring their effects when incorporated into individual proteins or at the cellular proteome-level. However, the impact of D-enantiomers of FAAs on the cellular metabolism remains largely unexplored and, to best of our knowledge, there are no reports on the biosynthesis of these compounds at chiral purity.

Alanine (Ala) represents an interesting model for studying the biosynthesis FAAs. The synthesis of Ala occurs through a single enzymatic step involving the reductive amination of pyruvate (Pyr). Owing to the small size of the methyl substituent in this amino acid, the proximity of the F substituent to the amino group allows strong electron-withdrawing effect of the halogen atom to influence the amino group[28], thus making Ala a particularly interesting case study. In terms of structure, glycine is the only amino acid that is simpler than Ala. However, fluorinated derivatives of glycine at the Cα position have been found to be unstable, undergoing rapid defluorination[29]. To date, examples on the bioproduction of 3-fluoroalanine (FAla) have been limited to the L-enantiomer [(R)-FAla], which can be obtained through the action of either L-Ala dehydrogenases or ω-transaminases on 3-fluoropyruvate (FPyr)[30–32]. The subsequent formation of the D-enantiomer [(S)-FAla] could theoretically be achieved through the action of an alanine racemase (Fig. 1a). However, this type of enzyme is inhibited by FAla[33], which contributes to the known antibacterial properties of some FAAs. An alternative approach could involve the use of *meso*-diaminopimelate dehydrogenases, enzymes that have garnered interest due to their broad substrate specificity and ability to produce target compounds with high enantiomeric excess[34,35]. *meso*-Diaminopimelate dehydrogenases have proven useful for the biosynthesis of different D-amino acids, including D-Ala[34] and D-phenylalanine[36]. Yet, there are no available reports of their activity on fluorinated derivatives of Pyr or in the production of FAAs—in contrast with the increasing need of synthetic tools to produce these building blocks towards establishing novel chemistries.

In this work, we explored the enzymatic production of mono- and tri-fluorinated Ala from F$_n$Pyr (with n = 1 or 3). To this end, we recombinantly produced an alanine dehydrogenase from *Vibrio proteolyticus* ([Vp]ALD) and a diaminopimelate dehydrogenase from *Symbiobacterium thermophilum* ([St]DAPDH). The kinetics of these two enzymes against FPyr and 3,3,3-trifluoropyruvate (F$_3$Pyr) were determined in vitro, together with the enantiomer distribution of the corresponding F$_n$Ala produced. An efficient NAD(P)H regeneration cycle, based on a NAD$^+$-dependent formate dehydrogenase (FDH) from *Pseudomonas* sp. 101 and an engineered derivative that exhibits increased specificity for NADP$^+$, was implemented to support high-yield bioproduction of FAla. This approach boosted the reaction yields to ~90% and 18 mM for the D-enantiomer and ~100% and 20 mM for the L-enantiomer of the FAA. Furthermore, an alanine racemase from *Streptomyces lavendulae* ([Sl]ALR), originally selected as a candidate for assembling these enzymatic cascades, was found to possess an unexpectedly high defluorinating activity on FAla *via* β-elimination. Taken together, the results in this article provide insights on the enzymatic synthesis of both a fluorinated D-amino acid and a trifluorinated version of Ala.

## Results and discussion
### Rationale behind enzyme selection, production and purification for establishing in vitro synthesis of fluorinated amino acids
In this study, the selection of enzymes was based on their potential to enhance the production of fluorinated alanine (FAla), with an initial round of biocatalyst selection informed by examples in the primary literature. Alanine dehydrogenase, derived from *Vibrio proteolyticus* ([Vp]ALD, a NAD$^+$-dependent dehydrogenase), was chosen due to its well-documented ability to maintain >70% substrate specificity for 3-fluoropyruvate (FPyr) in comparison to its non-fluorinated analogue, pyruvate (Pyr)[31]. The diaminopimelate dehydrogenase from *Symbiobacterium thermophilum* ([St]DAPDH, a NADP$^+$-dependent enzyme), on the other hand, was selected based on its known substrate promiscuity[34]. The Ala racemase of *Streptomyces lavendulae* ([Sl]ALR) was picked owing to its reported resistance to inactivation by D-cycloserine (4-amino-3-isoxazolidinone)[37]—a molecule that, similarly to FAla, can inhibit ALR by forming an adduct with the pyridoxal 5'-phosphate (PLP) cofactor and a key catalytic lysine residue in the active site[38]. The particular architecture of the active site of [Sl]ALR led to the hypothesis that this enzyme variant should be resistant to D-cycloserine inhibition, hence, it was hypothesized that this ALR variant might also exhibit increased tolerance towards FAla, an essential requirement in this study (Fig. 1a). Additionally, an Ala racemase from *Escherichia coli* ([Ec]ALR)

**Fig. 1 | Main reactions considered in this study. a** Predicted two-step synthesis of (S)-F$_n$-Ala by the sequential action of ALD and ALR. **b** One-step synthesis of (S)-F$_n$-Ala catalyzed by DAPDH. **c** FPyr regeneration by DAAO. **d** NAD(P)H regeneration by formate dehydrogenase.

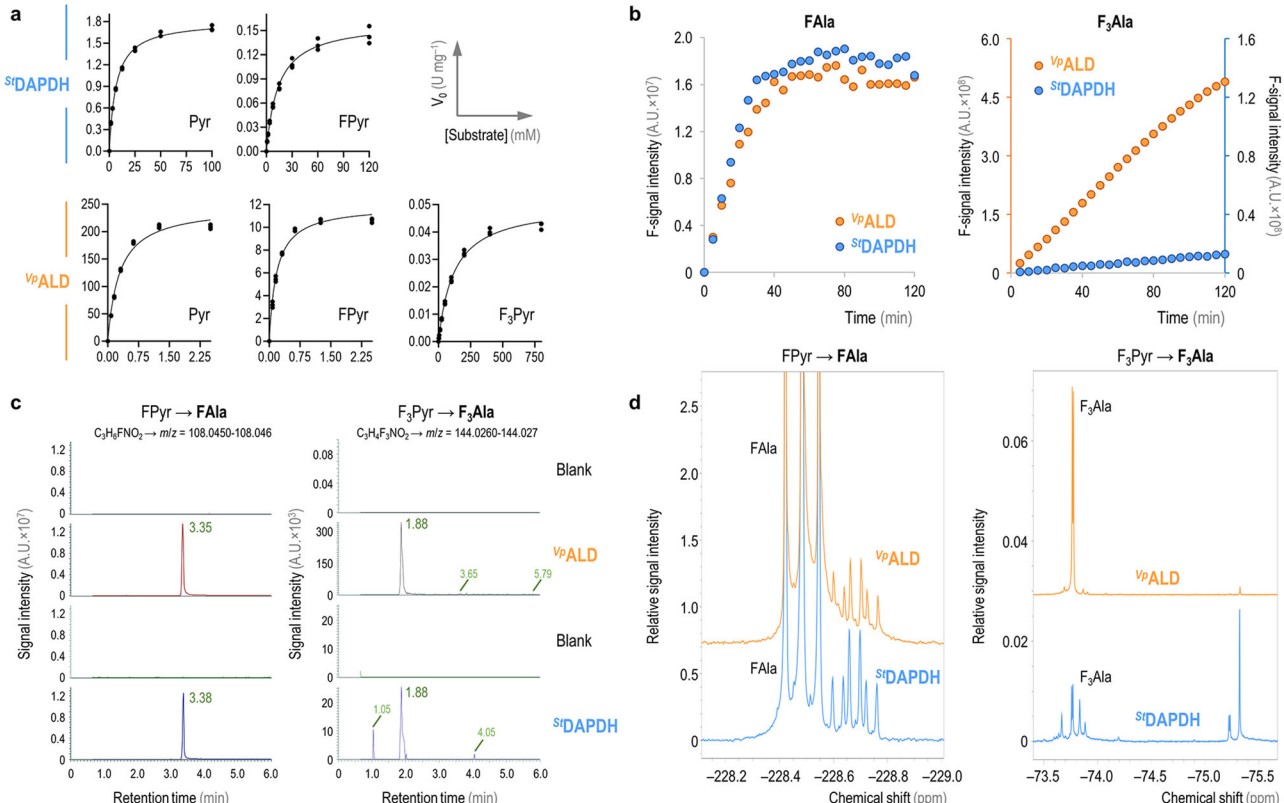

**Fig. 2 | Evaluating the activity of $^{St}$DAPDH and $^{Vp}$ALD in vitro. a** Kinetic data for the two enzymes assayed against Pyr, FPyr and $F_3$Pyr, with fitting to the canonical Michaelis-Menten equation. Kinetics were determined from three independent experiments; individual slope values are indicated in the plots. **b** Time-resolved $^{19}$F-NMR monitoring of FAla and $F_3$Ala production. The plots show the integration of the product peak over time for both $^{Vp}$ALD and $^{St}$DAPDH; due to the disparity on the levels of $F_3$Ala formation with the later enzyme, a secondary *y*-axis is shown (identified with the same color code as per the enzyme assayed). *A.U.*, arbitrary units.

**c** LC-MS spectra indicating the peaks with the *m/z* signal predicted for FAla and $F_3$Ala. The results shown for $^{Vp}$ALD and $^{St}$DAPDH are compared to the corresponding blank assay with no added enzyme. *A.U.*, arbitrary units. **d** Representative $^{19}$F-NMR spectra for the assays producing FAla and $F_3$Ala using the corresponding enzymes and substrates. The plots display a zoom-in of the chemical shifts predicted for each of the fluorinated products. The spectrum for $F_3$Ala formation by $^{St}$DAPDH was acquired using 4 times more scans than for $^{Vp}$ALD in order to increase the signal intensity.

was adopted as a reference, since its inhibition by FAla has been previously characterized[39]. Finally, a NAD$^+$-dependent formate dehydrogenase (FDH) from *Pseudomonas* sp. 101 (NAD-$^{Pse}$FDH) and an engineered variant with increased NADP$^+$ preference[40] were incorporated in the designs towards efficient regeneration of NAD(P)H, cofactors required by both $^{Vp}$ALD and $^{St}$DAPDH. All the enzymes were successfully produced in *E. coli* BL21(DE3) from the corresponding synthetic DNA fragments and purified to homogeneity through one-step Ni$^{2+}$-nitriloacetic (NTA) affinity chromatography. SDS-PAGE analysis (Supplementary Fig. S1) showed that all the proteins were isolated with a high degree of purity.

**Synthesis of FAla and $F_3$Ala from the corresponding pyruvate substrates by two dehydrogenases from *Vibrio proteolyticus* and *Symbiobacterium thermophilum***

The initial observation that alanine dehydrogenases (ALD) are capable of producing (*R*)-FAla through the reductive amination of FPyr was documented in the seminal work of ref. 30. Notwithstanding, this study predominantly focused on the affinity of this enzyme for halogenated substrates, yet a comprehensive kinetic characterization of $^{Vp}$ALD was missing not only in the original report but also in subsequent studies employing this enzyme as a biocatalyst[31,32,41]. Similarly, the broad substrate specificity of $^{St}$DAPDH has been underscored in the report by ref. 34, yet experimental assays specifically targeting FPyr have not been conducted for this enzyme or for any other biocatalyst within the *meso*-diaminopimelate dehydrogenase family. Furthermore, there is a dearth of studies exploring the enzymatic conversion of $F_3$Pyr into the corresponding trifluorinated Ala —a new-to-Nature building block with potential for a range of biocatalysis

applications[42]. To bridge this knowledge gap, particularly regarding the production of FAla *via* reductive amination, and to evaluate the potential application of this process in vivo, an in-depth analysis of the kinetics of both enzymes was executed in vitro. The experimental design involved using Pyr, FPyr and $F_3$Pyr as substrates. The enzymatic activities were quantified through continuous spectrophotometric monitoring of NAD(P)H oxidation according to the enzymatic reaction in the reductive substrate amination direction, as indicated for pathways involving either ALD (Fig. 1a) or DAPDH (Fig. 1b). The kinetic data were analyzed through the canonical Michaelis-Menten model (Fig. 2a).

$^{Vp}$ALD showed a similar affinity for the non-fluorinated and the monofluorinated Pyr substrate ($Km = 0.29$ and $0.18$ mM, respectively; Table 1), but the presence of F mediated a dramatic decrease in the dehydrogenase activity (i.e., only ca. 5% of the $V_{max}$ was retained when using FPyr as substrate). This result seems to contradict the >70% specific activity on FPyr previously reported for this enzyme[31], probably attributable to differences in the assay setup (e.g., substrate concentration). The $V_{max}$ value attained in our present study, however, is close to the ~6% reported for the ALD enzyme from *Helicobacter aurati* when acting on fluorinated Pyr[32]. One way or the other, $k_{cat}/Km$ was reduced by >10-fold (from ca. 597 to 46 s$^{-1}$ mM$^{-1}$) when FPyr was used as the substrate. $F_3$Pyr, on the other hand, had a strong detrimental effect on both the substrate affinity ($Km = 121$ mM) and the activity (ca. 0.05 U mg$^{-1}$) of $^{Vp}$ALD (Table 1). The strong negative impact on the enzyme activity might be due to the polarizing effect of the three F atoms (in the CF$_3$ group), which are expected to substantially modify the formal distribution of charges in their molecular surroundings[43]. Conversely, the loss of substrate affinity might seem more surprising, since

**Table 1 | In vitro kinetic characterization of $^{Vp}$ALD and $^{St}$DAPDH**

| Enzyme | Substrate | Kinetic parameter | | | |
|---|---|---|---|---|---|
| | | $Km$ (mM) | $V_{max}$ (U mg$^{-1}$) | $k_{cat}$ (s$^{-1}$) | $k_{cat}/Km$ (s$^{-1}$ mM$^{-1}$) |
| $^{Vp}$ALD | Pyr | 0.29 ± 0.04 | 247 ± 11 | 173 ± 8 | 596.6 |
| | FPyr | 0.18 ± 0.02 | 11.9 ± 0.4 | 8.3 ± 0.3 | 46.1 |
| | F$_3$Pyr | 121 ± 8 | 0.051 ± 0.001 | 3.53×10$^{-2}$ ± 8×10$^{-4}$ | 2.9 × 10$^{-4}$ |
| $^{St}$DAPDH | Pyr | 6.7 ± 0.3 | 1.81 ± 0.02 | 1.05 ± 0.01 | 0.16 |
| | FPyr | 14.0 ± 0.6 | 0.161 ± 0.002 | 0.093 ± 0.001 | 6.6 × 10$^{-3}$ |
| | F$_3$Pyr | N.D. | N.D. | N.D. | N.D. |

The parameters reported in the table correspond to regression coefficients obtained by fitting the mean values from three independent experiments to the Michaelis-Menten model ± standard deviation of the regression coefficient estimates. *N.D.*, not detected.

the small size of F is not predicted to alter the intramolecular structure significantly or cause steric hindrances at the catalytic pocket[9]. However, the analysis of the crystal structure of the ALD enzyme of *Mycobacterium tuberculosis* indicates that the stabilization of Pyr in the active site depends on the formation of specific hydrogen bonds and charge interactions through three conserved amino acid residues[44], a conformational architecture that might be disturbed by the polarizing effect of the three F substituents in F$_3$Pyr. The catalytic residues involved in these interactions (Arg15, Lys75 and His96) were confirmed to play the same stabilizing role in the $^{Gk3448}$ALD enzyme of *Geobacillus kaustophilus*[45], and they are all conserved in $^{Vp}$ALD (Supplementary Fig. S2). Prompted by these results, we also analyzed the catalytic performance of $^{St}$DAPDH under the same reaction conditions (Fig. 2a). The affinity of $^{St}$DAPDH for the non-fluorinated, native substrate was ca. 23-fold lower than that of $^{Vp}$ALD; the $k_{cat}$ values were similarly lower for the dehydrogenase of *S. thermophilum*. In general, the trend observed in the $Km$ and $V_{max}$ values evaluated in the presence of fluorinated substrates was similar to the results obtained for the $^{Vp}$ALD enzyme. The effect of FPyr was especially noticeable at the activity level, while the affinity was kept within the same order of magnitude. The $k_{cat}/Km$ was reduced by ca. 1000-fold in the presence of FPyr (ca. $7 \times 10^{-3}$ s$^{-1}$ mM$^{-1}$). Moreover, in this case, the reduction of both affinity and activity when using F$_3$Pyr as the substrate was too strong to allow for the determination of enzyme kinetic parameters (Table 1).

To facilitate a direct and quantitative assessment of the formation of the target fluorinated products, serving as a supplementary method to the in vitro assays (Fig. 2a), individual reactions were prepared under the same conditions as described above and analyzed by both $^{19}$F-NMR (Fig. 2b, d) and high-resolution LC-MS(MS) (Fig. 2c). In the case of the $^{19}$F-NMR assays, the dynamics of both substrate depletion and product generation in the reactions were continuously monitored in real-time. This combined analytical approach provided a deeper understanding of the reaction kinetics and enabled the direct visualization of the transformation of the substrates within the reaction milieu. Both FPyr and F$_3$Pyr could be easily identified by $^{19}$F-NMR (Supplementary Fig. S3A and S3B, respectively). In general, the chemical shifts and the multiplet patterns observed in these samples matched the expected signals for both FAla and F$_3$Ala, regardless of the selected enzyme ($^{Vp}$ALD or $^{St}$DAPDH, Fig. 2d). FAla formation, evaluated as the time-resolved intensity of the F-signal in the samples, reached a plateau around 40 min of incubation for both enzymes (Fig. 2b). Conversely, the transformation of F$_3$Pyr followed a linear kinetic over 2 h, with a significantly lower conversion rate mediated by the action of $^{St}$DAPDH (Fig. 2b), suggesting that the lower rates observed for this substrate prevented the reactions to reach equilibrium within the duration of the assay. These results recapitulate the experimental observations in the assays where product formation was inferred from the rates of NADH oxidation (Table 1).

To further verify the chemical identity of the expected FAAs, the samples were also submitted to LC-MS(MS) analysis in positive-heated electrospray ionization (HESI) mode (Fig. 2c). While no significant signal could be detected in the blank (control) experiments, the reaction samples

displayed signals with an experimental *m/z* fitting the predicted values for FAla and F$_3$Ala—for FAla (C$_3$H$_6$FNO$_2$), *m/z* = 108.045-108.046 and for F$_3$Ala (C$_3$H$_6$F$_3$NO$_2$), *m/z* = 144.026-144.027 (Fig. 2c). The predicted masses for both enzymes and substrates were likewise confirmed in these assays. An MS(MS) validation was successfully carried out for every condition except for F$_3$Ala production by $^{St}$DAPDH, due to limited signal intensity. The full MS(MS) spectra for the conversion of FPyr into FAla by both $^{Vp}$ALD and $^{St}$DAPDH are presented in Supplementary Fig. S4A and S4B, respectively; while the MS(MS) spectrum for the transformation of F$_3$Pyr by $^{Vp}$ALD is shown in Supplementary Fig. S5. The $^{19}$F-NMR spectra further confirmed the identity of the target fluorinated products, with relative intensities considerably higher for FAla than F$_3$Ala (Fig. 2d). Accordingly, the spectra for the trifluorinated product were acquired using 4 times more scans to increase the signal intensity. Interestingly, the analysis of the performance of $^{St}$DAPDH against F$_3$Pyr as a substrate revealed the formation of side products with signal intensities within the range of (or even higher than) that expected for F$_3$Ala (Fig. 2d). Both the chemical shift and the singlet nature of these signals are suggestive of the presence of a –CF$_3$ group. In addition, no signal of free fluoride was detected by $^{19}$F-NMR (Supplementary Fig. S6) indicating that the CF$_3$ group was present in all the fluorochemicals of the reaction mixtures. Importantly, the fluorinated side product in the region of –75.4 ppm seemed to be also present in trace amounts for the conversion catalyzed by $^{Vp}$ALD. These unidentified compounds are likely the result of an alternative catalytic fate in the active sites of $^{Vp}$ALD and $^{St}$DAPDH[46,47].

## An alanine racemase from *Streptomyces lavendulae* displays a secondary dehalogenation activity on fluorinated amino acid analogues

As indicated in the previous section, $^{Vp}$ALD had the best catalytic efficiency for the reductive amination of FPyr, with a $k_{cat}/Km$ ca. 7000-fold higher than $^{St}$DAPDH (Table 1). However, (*R*)-FAla is the predicted product for the $^{Vp}$ALD activity, which requires the action of a racemase to convert the enantiomer product into (*S*)-FAla. Hence, an experimental screening was established to investigate if $^{Sl}$ALR is able to both escape the inhibition mediated by FAla while catalyzing the formation of the desired enantiomer. As a first step, the $^{Sl}$ALR and $^{Ec}$ALR racemases were assayed against their natural substrate, L-Ala, by coupling the reaction to the Pyr-dependent formation of L-lactate (Supplementary Fig. S7). The reaction was monitored by following the rate of NAD$^+$ generation, which indicated that both $^{Sl}$ALR and $^{Ec}$ALR were active under these conditions (Supplementary Fig. S8a and S8b, respectively). Next, a similar spectrophotometric assay was designed to couple the activity of ALR with the oxidation of NADH in a different configuration (Fig. 3a). In this case, the activity of $^{Vp}$ALD on FPyr was combined with $^{Sl}$ALR and a D-amino acid oxidase (DAAO, a commercially-available DAAO from porcine kidney). DAAO displays specific activity on the D-enantiomer, and this enzyme was expected to regenerate FPyr while producing H$_2$O$_2$ as byproduct of the deamination reaction. Owing to the formation of this byproduct, which could potentially inhibit the enzymes if the peroxide concentration increases above a threshold level, a catalase from bovine liver was added to the reaction

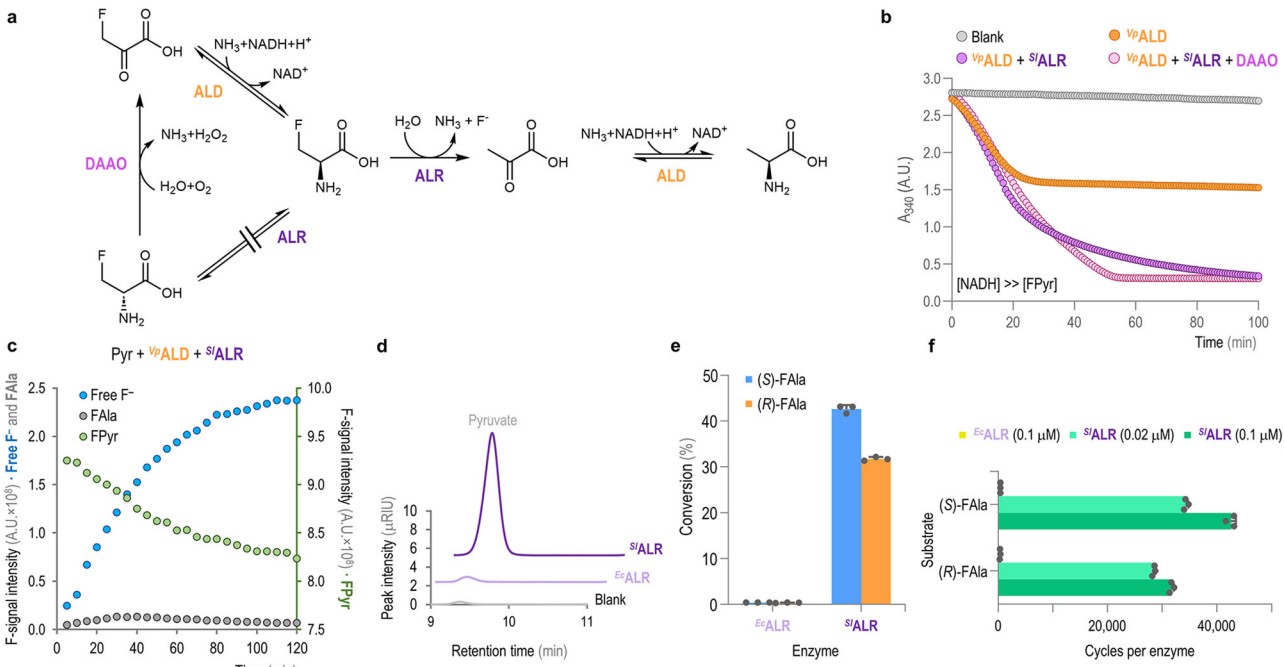

**Fig. 3 | Exploring the dehalogenation activity of $^{Sl}$ALR. a** The scheme depicts the possible pathways for the synthesis of an ALD substrate, which can be formed through β-elimination of the F atom or *via* the action of a racemase upon the addition of DAAO. **b** Spectrophotometric monitoring of NADH oxidation using FPyr as limiting substrate with different enzyme combinations. The graphs represent mean values and the error bars correspond to standard deviations from three independent experiments. N.B., error bars are smaller than the corresponding mean data point. *A.U.*, arbitrary units. **c** Time-resolved $^{19}$F-NMR monitoring of FAla formation, FPyr consumption and release of free fluoride (F⁻). The axes are colored according to the chemical species measured in each case. *A.U.*, arbitrary units.

**d** Representative example of the HPLC profiles for assays incubated with either $^{Sl}$ALR or $^{Ec}$ALR and the blank for the same reaction conditions. The plot shows the region encompassing the characteristic retention time of Pyr. *μRIU*, refractive index units × 10⁻⁶. **e** Substrate conversion per enantiomer [(S)- and (R)-FAla], determined from the release of Pyr as compared to the initial amount of FAla in the assays. **f** Catalytic cycles per unit of enzyme for both substrates and different enzyme amounts. In panels **e** and **f**, the graphs represent mean values and the error bars correspond to standard deviations from three independent experiments; individual data points are indicated in the plots.

mixture. In these tests, NADH was added in excess (1 mM) in comparison to the FPyr concentration (0.5 mM) as a strategy to study the functionality of $^{Sl}$ALR through the formation of NAD⁺ while preventing any cofactor limitation. The underlying reasoning was that the reaction could only proceed to completion (i.e., NADH exhaustion) only if the DAAO activity effectively replenished FPyr (Fig. 1c). Control experiments were prepared with either no enzymes added, $^{Vp}$ALD alone and also combining $^{Vp}$ALD and $^{Sl}$ALR but without DAAO. The latter assay was set to account for the potential β-elimination activity of the ALR enzyme, which would lead to the release of non-fluorinated Pyr, thus creating an alternative regeneration cycle (Fig. 3a). Equivalent reaction mixtures, using an excess of FPyr (5 mM), were prepared as additional controls (Supplementary Fig. S9). In these reactions, all NADH seemed to be consumed at similar rates in <40 min across the experimental conditions tested.

Surprisingly, in the reactions where FPyr was the limiting substrate, the combination of $^{Vp}$ALD and $^{Sl}$ALR was sufficient to reach similar NADH oxidation levels as those observed in assays containing $^{Vp}$ALD, $^{Sl}$ALR and DAOO (Fig. 3b). As expected, NAD⁺ formation was rather limited in assays containing $^{Vp}$ALD alone (ca. half of the NADH was oxidized over the first 30 min of the test, indicative of the conversion of FPyr into FAla without any further transformation taking place). These results suggested that the β-elimination activity of $^{Sl}$ALR had to be much higher than the levels reported for its orthologue from *E. coli*, an ALR enzyme thoroughly characterized in vitro[39,48]. To further substantiate this hypothesis, the dynamics of the fluorinated substrate (FPyr) and potential product(s) thereof were monitored by $^{19}$F-NMR in assays combining $^{Vp}$ALD and $^{Sl}$ALR (Fig. 3c and Supplementary Fig. S10). This time-resolved experiment revealed a steady release of free fluoride (F⁻) from the substrate that essentially mirrored the

decrease in the FPyr-related F-signal, while the amount of FAla remained at background levels throughout the 2 h assay.

To explore the extent of the defluorination capacities of $^{Sl}$ALR, a direct assay was prepared by testing the activity of this enzyme against the pure enantiomers, i.e., (R)-FAla and (S)-FAla. Equivalent reactions were prepared with $^{Ec}$ALR to be used as a reference, on the basis of the well-characterized inhibition of this enzyme in the presence of either FAla enantiomer. In these assays, the β-elimination activity of the ALRs was evaluated through the quantification of the released Pyr by HPLC, which is stoichiometric with respect to the formed F⁻ *via* β-elimination. The HPLC profile of these reactions indicated that Pyr was indeed formed in the reactions containing $^{Sl}$ALR, while residual amounts of the non-fluorinated product could be detected in the assays with $^{Ec}$ALR (Fig. 3d). Furthermore, (R)-FAla and (S)-FAla were transformed by $^{Sl}$ALR into Pyr at ca. 32% and 43% conversion yields, respectively (Fig. 3e). Conversely, $^{Ec}$ALR had a very low level of β-elimination activity (paired to the inhibitory effect of fluorinated substrates), and only <0.5% substrate conversion was detected for both FAAs.

The number of catalytic cycles per enzyme unit (Ψ) was also explored for the two racemase enzymes under study (Fig. 3f). Two biocatalyst concentrations (0.02 and 0.1 μM) were used in these assays in order to cover a 5-fold range of enzyme/substrate ratios, and the Ψ values were determined as molecules of product per molecule of enzyme. Under these assay conditions, Pyr formation was below the experimental limit of detection when $^{Ec}$ALR was used at 0.02 μM. When the biocatalyst concentration was increased to 0.1 μM, $^{Ec}$ALR mediated Ψ = 400 cycles for (S)-FAla and Ψ = 300 cycles for (R)-Fala (Fig. 3f). $^{Sl}$ALR, on the other hand, reached Ψ = 34,000−42,000 cycles for (S)-FAla and Ψ = 28,000−31,000 cycles for

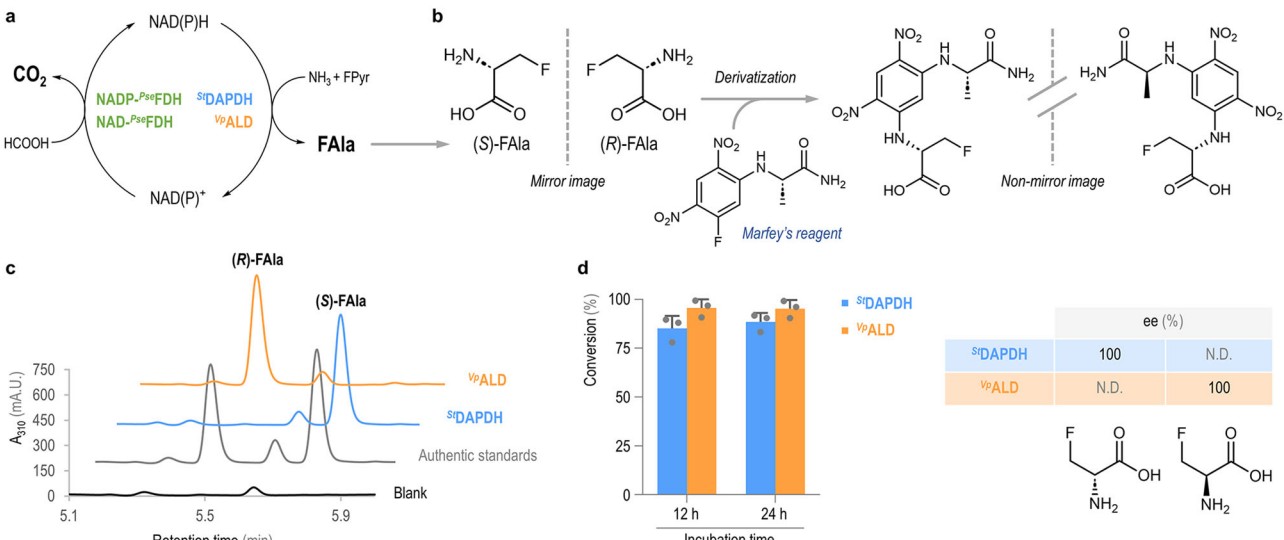

**Fig. 4 | Enzymatic production of FAla in vitro and a workflow to determine the enantiomeric excess. a** The scheme indicates the enzymatic cascade assembled for FAla synthesis coupled with the regeneration of the reduced cofactor [NAD(P)H]. The enzyme pair $^{St}$DAPDH and NADP-$^{Pse}$FDH uses FPyr and formate as substrates; FPyr is converted into (S)-FAla by $^{St}$DAPDH, whereas NADPH is regenerated by formate oxidation to $CO_2$. The equivalent reactions are catalyzed by the enzyme pair $^{Vp}$ALD and NAD-$^{Pse}$FDH, in this case forming (R)-FAla and providing NADH. **b** Chiral purity is assessed after the reaction mixture is derivatized with the Marfey's reagent (1-fluoro-2,4-dinitrophenyl-5-L-alanine amide, FDAA), which converts the

FAla enantiomers into non-enantiomeric derivatives that can be separated by reversed-phase chromatography. **c** Representative HPLC profiles showing a reaction blank, a mixture of authentic (S)-FAla and (R)-FAla standards, a sample of the reaction containing $^{St}$DAPDH and a sample with $^{Vp}$ALD. *mA.U.*, arbitrary units $\times$ $10^{-3}$. **d** Quantification of the FAla formation in terms of FPyr conversion and enantiomeric excess, ee, for $^{Vp}$ALD and $^{St}$DAPDH. The graph represents mean values and the error bars correspond to standard deviations from three independent experiments; individual data points are indicated in the plot. *N.D.*, not detected.

(R)-FAla when the racemase was assayed at 0.02 and 0.1 μM, respectively (Fig. 3f). Therefore, $^{Sl}$ALR had an overall 100-fold increased capacity of mediating FPyr defluorination as compared to $^{Ec}$ALR. In this sense, the number of β-elimination cycles prior irreversible inactivation by FAla (both enantiomers) has been experimentally determined for $^{Ec}$ALR[39] and other microbial racemases from *Salmonella*[38,49] and *Pseudomonas* species[50]. In all these documented cases, Ψ = 800 cycles (or even lower), which is substantially lower than our experimental results for $^{Sl}$ALR on this FAA. Although the in vitro assay adopted throughout this study was merely designed to measure the number of catalytic cycles per unit of enzyme under selected experimental conditions, our results would indicate that $^{Sl}$ALR exhibits an unprecedent defluorination capacity. This observation may have a functional connection to the fact that *Streptomyces* species are known to host reactions for fluorometabolite biosynthesis[51], which require exquisite biochemical[52] and physiological adaptations[53] to avoid toxicity. Based on the results described in this section, $^{Sl}$ALR was deemed unsuitable to racemize (R)-FAla as required to produce D-FAAs—but it might be repurposed as a biodehalogenation tool in future research.

**In vitro NAD(P)H regeneration supports continuous production of FAla enantiomers with high yields and purity**

The main technical limitation for the enzymatic production of FAla from FPyr is the high cost of the redox cofactor. Hence, cofactor regeneration strategies were contemplated in an attempt to improve the yield of the target FAA in a cost-effective fashion. In this sense, energy- and redox-cofactor regeneration cycles have been successfully included as part of enzymatic cascades in vitro to boost catalytic efficiency and product formation[54–56]. One of the first approaches implemented to solve this stoichiometric bottleneck for FAA synthesis was proposed by ref. 30, and it consisted of coupling NADH regeneration to formate oxidation by means of a FDH enzyme from *Saccharomyces cerevisiae*. Not only is formate a low-cost, readily available additive, but the product of the auxiliary reaction is $CO_2$, which is spontaneously eliminated from the reaction mixture by evaporation. While this setup works reasonably well with NADH, NADP$^+$ reduction by FDH is notoriously difficult. Previous attempts to modify the

cofactor specificity of these dehydrogenases came at the cost of a dramatic loss of affinity on the carbon substrate ($Km$ = 1000 mM for formate)[57]. In this work, the FDH from *Pseudomonas* sp. 101 was selected to support redox cofactor cycling (Fig. 1d). Importantly, an engineered version of this enzyme can efficiently use NADP$^+$ as the cofactor while retaining ~15% of the catalytic efficiency of the wild-type dehydrogenase against formate[40]. Thus, NAD-$^{Pse}$FDH (i.e., the original enzyme from *Pseudomonas* sp. 101) was used to regenerate NADH to support the synthesis of (R)-FAla from FPyr, catalyzed by $^{Vp}$ADLH (Fig. 4a). Similarly, NADP-$^{Pse}$FDH was adopted to regenerate NADPH for the production of (S)-FAla, catalyzed by $^{St}$DADPH (Fig. 4a). A separate set of in vitro assays confirmed the high specificity for either cofactor of the purified NAD-$^{Pse}$FDH and NADP-$^{Pse}$FDH enzymes (Supplementary Fig. S11a and S11b, respectively). Both NH$_4$Cl and formate were added to these cascade reactions at high concentration (200 mM) to shift the equilibrium towards the reductive amination. The corresponding reactions were analyzed in terms of total product formation and enantiomeric distribution. In this case, a derivatization step with the commercially-available Marfey's reagent (1-fluoro-2,4-dinitrophenyl-5-L-alanine amide, FDAA) was incorporated into the experimental design to enable the separation of the two FAla enantiomers by reversed-phase chromatography (Fig. 4b). Calibration curves, prepared with the authentic (R)- and (S)-FAla standards treated with the Marfey's reagent (Supplementary Fig. S12a and S12b, respectively), were run in parallel for each set of experiments.

Analysis of the HPLC profiles indicated that this methodology allowed for the determination of chiral purity (Fig. 4c), thus the R- and the S-enantiomers eluted as well defined and separated peaks. The chromatogram also displayed two minor peaks in the region of interest, one at 5.6 min and another at 5.3 min, but they were also detected at the controls (Supplementary Fig. S13) and can be attributed to the excess of derivatization reagent[58,59]. Based on the chromatographic analysis both $^{Vp}$ALD and $^{St}$DAPDH were determined to have a specific pattern of product formation, mediating the selective synthesis of (R)-FAla and (S)-FAla, respectively. High-conversion yields were detected for both enzymes, with ~100% and ~90% conversion of FPyr into the target FAA products by $^{Vp}$ALD and

$^{St}$DAPDH, respectively (Fig. 4d). These product yield values were similarly high when comparing reactions incubated for either 12 or 24 h, suggesting that the excess of NH$_4$Cl and formate was sufficient to keep the reaction equilibrium shifted in the reductive amination direction. Moreover, the enantiomeric excess was 100% for both enzymes (Fig. 4d), with no significant detection of the alternative enantiomer form of the FAA—underscoring the very selective nature of the reaction catalyzed by either dehydrogenase. Under these conditions and upon an incubation of 12 h, (R)-FAla was produced at 20 mM in the enzymatic cascade containing $^{Vp}$ALD, whereas the concentration of (S)-FAla reached 18 mM when $^{St}$DADPH was used as the biocatalyst. Analyzing the process in terms of units of product per unit of catalytic center (total turnover number, TTN) and units of product per unit of catalytic center and unit of time (turnover frequency, TOF), $^{Vp}$ALD reached 4000 and 330 h$^{-1}$, respectively, whereas the values for $^{St}$DAPDH were 360 and 30 h$^{-1}$.

The conversion values highlight the functionality of the FDH-based regeneration system, without which the maximal production levels would be constrained by the amount of reduced cofactor added (1 mM). Furthermore, the FAla yields on substrate obtained in this study rank among the highest reported for FAAs, while the specific formation of the D-enantiomer had not been reported thus far. While these results were somewhat within the range expected for $^{Vp}$ADLH[30], the high performance of $^{St}$DADPH underscores the potential of this enzyme (and other diaminopimelate dehydrogenases) as a promising candidate for supporting the enzymatic production of halogenated D-amino acids.

## Conclusions

The present study provides new insights on the enzymatic production of fluorinated versions of Ala through reductive amination. As the biocatalysis toolbox for the production of alternative building blocks, e.g., NCAAs, continues to expand[60–62], approaches for the selective production of FAA enantiomers (especially, Ala[63]) are increasingly needed. In this work, a kinetic characterization of $^{St}$DADPH against FPyr and $^{Vp}$ALD against both FPyr and F$_3$Pyr showed that the monofluorinated substrate decreased the reaction rate with little effect on the enzyme affinity, whereas the trifluorinated version of Pyr exhibited a strong detrimental impact on both affinity and reaction velocity. The production of FAla was combined with the simultaneous regeneration of the reduced cofactor through the oxidation of inexpensive formate by NAD(P)-$^{Pse}$FDH. The implementation of this efficient regeneration system led to high conversion yields and enantiomeric purity for both (S)-FAla and (R)-FAla. $^{Sl}$ALR was determined to be an enzyme with an unexpected high efficiency in dehalogenating FAla through β-elimination, suggesting potential applications in the field of bioremediation[64–66]. Taken together, our results provide novel approaches to the synthesis of non-canonical building blocks for life[24], which constitutes a decisive step to engineering living cells[67–70] with alternative lifestyles and functions[71].

## Methods

### Chemicals and reagents

(S)-3-Fluoroalanine and (R)-3-fluoroalanine were purchased from BLD Pharmatech Ltd. (Shanghai, China). Trifluoropyruvic acid was acquired from Fluorochem Ltd. (Glossop, UK), isopropyl-β-D-1-thiogalactopyranoside (IPTG) was purchased from Biosynth AG (Staad, Switzerland). All other reagents were acquired from Sigma-Aldrich Co. (St. Louis, MO, USA) unless otherwise specified.

### DNA synthesis, protein production and purification

The list of proteins produced and purified in this study comprise an Ala dehydrogenase from *Vibrio proteolyticus* ($^{Vp}$ALD, UniProt ID O85596), a diaminopimelate dehydrogenase from *Symbiobacterium thermophilum* ($^{St}$DAPDH, UniProt ID Q67PI3), an Ala racemase from *Streptomyces lavendulae* ($^{Sl}$ALR, UniProt ID Q65YW7), an Ala racemase from *Escherichia coli* ($^{Ec}$ALR, UniProt ID P0A6B4), a NAD$^+$-dependent formate dehydrogenase (FDH) from *Pseudomonas* sp. 101 (i.e., NAD-$^{Pse}$FDH) and

its site-specific engineered variant (originally named FDH V9)[40], which is characterized by a shifted specificity towards NADP$^+$ as a cofactor[72]. For the sake of simplicity, the latter was termed NADP-$^{Pse}$FDH throughout this study. The gene fragments encoding $^{Vp}$ALD, $^{St}$DAPDH, $^{Sl}$ALR and $^{Ec}$ALR were codon-optimized for expression in *E. coli* K12 (and other Gram-negative bacteria) as explained elsewhere[73] and synthesized de novo by Twist Bioscience (San Francisco, CA, USA). Plasmids pZ-ASL[40] carrying either the coding sequence of the parental NAD-$^{Pse}$FDH or its engineered version were kindly provided by the Bar-Even group (Max Planck Institute of Molecular Plant Physiology, Golm, Germany). The selected genes were incorporated into a modified pET28a(+) vector (Novagen$^{TM}$, Sigma-Aldrich Co.) encoding the tobacco etch virus (TEV) protease-cleavage site instead of the conventional recognition motif for thrombin protease; the relevant coding sequences were amplified by PCR from the synthetic DNA and the final constructs were obtained through uracil excision (*USER*) cloning[74]. To this end, USER-primers were designed using the AMUSER tool[75]; the native *START* codon of the genes indicated above was removed and the DNA sequence was modified to encode a protein carrying a *N*-terminal His$_{6\times}$-tag followed by the TEV site. The codon-optimized gene sequences and the oligonucleotides needed for USER assembly are listed in Supplementary Tables S1 and S2; the cloning procedures used in this study followed well-established protocols[70,76–78]. Chemically-competent *E. coli* DH5α λ*pir* cells[79] were used for routinary gene cloning and plasmid construction, and the resulting constructs were sequence-verified prior to be transferred into the expression host *E. coli* BL21(DE3) [F$^-$ λ$^-$ *ompT hsdSB*(r$_B^-$, m$_B^-$) *gal dcm* (DE3); Thermo Fisher Co., San Jose, CA, USA] for protein production.

Protein production experiments started with the preparation of 10-mL precultures by inoculating single colonies of the corresponding recombinant *E. coli* BL21(DE3) strains. Precultures were prepared in 2×YT medium[80] supplemented with 50 µg mL$^{-1}$ kanamycin and incubated overnight at 37 °C with agitation at 200 rpm. Then, a 5-mL aliquot of the precultures was used to start 500-mL cultures set in 2-L baffled Erlenmeyer flasks, using 2×YT medium supplemented with 50 µg mL$^{-1}$ kanamycin. These cultures were incubated under the same conditions indicated above until an optical density at 600 nm (OD$_{600}$) of 0.5−0.7 was reached, whereupon the cultures were cooled down to 4 °C with no further shaking. Gene expression was induced by adding IPTG to the cultures at a final concentration of 0.4 mM. The induced cultures were incubated at 20 °C and 200 rpm for 18 h, then harvested by centrifugation (4000 × *g*, 20 min, 4 °C) and stored at –20 °C until further processing. Cell lysis was done by sonication, resuspending the frozen bacterial pellets in 10 mL of buffer A (20 mM sodium phosphate pH = 7.5, 300 mM NaCl and 20 mM imidazole) and then submitting the suspension to 2 rounds of ultrasound treatment using a Vibra-Cell$^{TM}$ instrument (model VCX130; Sonics & Materials Co., Newtown, CT, USA) equipped with a 6-mm probe (ref. 630-0422). The procedure consisted of two 7 min-series of sonication at 50% amplitude through 30 s/30 s ON/OFF cycles while keeping the samples on ice to prevent overheating. The lysates were then treated with 2.5 U mL$^{-1}$ Pierce$^{TM}$ universal nuclease for cell lysis (Thermo Fisher Scientific Co.) at room temperature with gentle shaking for 30 min, followed by 20 min centrifugation at 12,000 × *g* and 4 °C to remove cell debris. The resulting supernatants were filtered through 0.2 µm-membranes and the His$_{6\times}$-tagged enzymes were purified by means of immobilized metal chelate affinity chromatography[81]. Purification was performed using 1 mL of Ni$^{2+}$-nitriloacetic acid (NTA) resin (HisPur$^{TM}$ Ni-NTA resin, Thermo Fisher Scientific Co.) loaded onto 10 mL-Pierce$^{TM}$ disposable columns. The resin was equilibrated in 10 mL of buffer A, then the corresponding clear lysate was passed through the resin, followed by a washing step with 20 mL of buffer A. The bound proteins were eluted by applying 4 mL of buffer B (20 mM sodium phosphate, pH = 7.5, 300 mM NaCl and 500 mM imidazole). The next step consisted of exchanging the buffer to C (20 mM sodium phosphate pH = 7.5, 300 mM NaCl and 1 mM EDTA) by centrifuging the purified enzyme preparations (4000 × *g*, 4 °C) using Amicon$^{TM}$ Ultra-15 centrifugal filters of 10 kDa-pore size (Merck-Milli-PoreSigma, Burlington, MA, USA). The absorbance at 280 nm (A$_{280}$) of purified enzyme preparations was measured in a NanoDrop$^{TM}$

2000 spectrophotometer (Thermo Fisher Scientific Co.) and the protein concentration was determined based on the respective theoretical molecular weight and the molar extinction coefficient ($\varepsilon_{280}$). The predicted amino-acid sequences of the isolated proteins, together with their calculated molar weight and molar extinction coefficient are displayed in Supplementary Table S3. The purity of the recombinant enzymes was validated by SDS-PAGE, and the enzyme preparations were stored at 4 °C until further use.

### Enzyme assays and assembly of in vitro enzyme cascades

Unless otherwise stated, the enzymatic activities were determined by following the oxidation of the NAD(P)H cofactor over time. Samples were prepared in triplicates in a final volume of 200 µL and vigorously mixed with the assay mixture. The absorbance at 340 nm ($A_{340}$) was monitored in 50 s-intervals at 30 °C in an *EPOCH2* microplate reader (BioTek Instruments, Winooski, VT, USA). Enzymatic activities were determined by comparison with the corresponding blank experiments, without any catalyst added. The composition of the different reaction mixtures included 50 mM sodium phosphate (pH = 8.0), 200 mM $NH_4Cl$ and 0.1 g $L^{-1}$ bovine serum albumin. The initial concentration of $F_nPyr$, enzyme and NAD(P)H was adjusted depending on the specific requirements of the assay. NADH and NADPH were added to assays containing $^{Vp}$ALD and $^{St}$DAPDH, respectively. Pyridoxal 5'-phosphate was added at 50 µM to assays including an ALR enzyme. Thus, the kinetics for $^{Vp}$ALD were determined using 0.5 mM NADH and a substrate range of $0-2.5$ mM for Pyr; $0-10$ mM for FPyr and $0-800$ mM for $F_3Pyr$. The kinetics for $^{St}$DAPDH were analyzed against Pyr ($0-100$ mM) and FPyr ($0-120$ mM) using 0.5 mM NADPH in all cases. Kinetic data were fit to the classical Michaelis-Menten kinetic model using SigmaPlot 15.0 (Systat Software Inc., San Jose, CA, USA). Reaction samples to be analyzed by LC-MS or NMR were prepared with an increased concentration of NAD(P)H of 2 mM in order to boost the final amount of fluorinated product(s).

In LC-MS analyses, the enzymatic production of FAla was assayed by using 10 mM FPyr and 0.05 µM $^{Vp}$ALD or 20 µM $^{St}$DAPDH, as indicated in the specific experiment. The biosynthesis of $F_3Ala$ was performed by mixing 100 mM $F_3Pyr$ with 8 µM $^{Vp}$ALD or 150 µM $^{St}$DAPDH. The reactions were incubated at 800 rpm and 30 °C for 2 h, before stopping the assay by adding methanol at a final concentration of 50% (v/v). In the case of $^{St}$DAPDH and when using $F_3Pyr$ as the substrate, the reaction time was extended to 20 h. For NMR analyses, the synthesis of FAla was carried out by using 10 mM FPyr and 0.05 µM $^{Vp}$ALD or 20 µM $^{St}$DAPDH. The enzymatic production of $F_3Ala$ was performed with 100 mM $F_3Pyr$ and 8 µM $^{Vp}$ALD or 150 µM $^{St}$DAPDH. In all cases, deuterated water ($D_2O$) was added at 10% (v/v). The reaction progression was monitored by recording the spectra over time at 30 °C as described in the text.

The functionality of $^{Sl}$ALR and $^{Ec}$ALR was first verified on their natural substrate L-Ala (Supplementary Method S1). An enzyme cascade was designed to assay the activity of $^{Sl}$ALR against (R)-FAla (Fig. 1c). $^{Vp}$ALD and $^{Sl}$ALR were combined (each at 0.05 µM) with 1 µ $mL^{-1}$ D-amino acid oxidase (DAAO) from porcine kidney (Merck product no. A5222) and 2 µ $mL^{-1}$ catalase from bovine liver (Merck-MilliPoreSigma, ref. C1345). NADH was added at a final concentration of 1 mM, and FPyr was used as a substrate either at 0.5 mM (limiting concentration compared to NADH) or 5 mM (excess concentration compared to NADH).

The ability of $^{Sl}$ALR and $^{Ec}$ALR to β-eliminate F atoms was assayed directly against (S)-FAla and (R)-FAla as the substrates. Two different amounts of enzyme, i.e., 0.02 µM and 0.1 µM, were added to 10 mM of each FAla isomer. Reactions were incubated at 30 °C and 800 rpm for 18 h and stopped by the end of the incubation through the prompt addition of formic acid at 5% (v/v). The elimination of F was analyzed by HPLC through the quantification of the released Pyr, stoichiometric with respect to $F^-$ formed. The number of catalytic cycles per unit of biocatalyst was calculated using two different amounts of enzymes (i.e., 0.02 µM and 0.1 µM) and comparing the amount of molecules of product measured with the number of molecules of ALR added to the assay.

### Continuous enzymatic production of FAla in vitro

Production of FAla from FPyr using either $^{Vp}$ALD or $^{St}$DAPDH was combined with (NAD/NADP)-$^{Pse}$FDH and formate as a regeneration system for NAD(P)H (Fig. 1d). First, the functionality of both $^{Pse}$FDH enzymes was verified on formate and the corresponding redox cofactors (Supplementary Method S2). For the continuous production of FAla, the substrate FPyr was added at 20 mM, whereas the initial concentration of formate and $NH_4Cl$ was set to 200 mM. For the production of (R)-FAla, $^{Vp}$ALD (5 µM) was combined with NAD-$^{Pse}$FDH (5 µM) and 1 mM NADH. For the production of (S)-FAla, $^{St}$DAPDH (50 µM) was combined with NADP-$^{Pse}$FDH (5 µM) and 1 mM NADPH. The reaction samples were incubated at 800 rpm and 30 °C for 12 and 24 h and stopped by freezing the reaction mixtures at $-80$ °C. In all cases, the production and the enantiomeric distribution of FAla on each condition were determined by HPLC. The HPLC analysis required a previous derivatization using the Marfey's reagent (1-fluoro-2,4-dinitrophenyl-5-L-alanine amide, FDAA), which reacts with primary amines allowing for the separation of amino acid enantiomers by reverse-phase chromatography. The reaction samples were diluted 1:25 prior to derivatization, which was performed by following the manufacturer's instructions (Thermo Fisher Scientific Co.).

### HPLC analysis

The β-elimination of F through the activity of either $^{Sl}$ALR or $^{Ec}$ALR against FAla was analyzed by HPLC, based on the formation of Pyr. The analysis was performed on a Dionex UltiMate$^{TM}$ 3000 HPLC system equipped with a RefractoMax 521 refractive index detector (IDEX Health & Science LLC, Oak Harbor, WA, USA). Samples were loaded onto an Aminex$^{TM}$ HPX-87X ion exclusion (300 × 7.8 mm) column (Bio-Rad Laboratories, Hercules, CA, USA) kept at 30 °C, and the mobile phase was 5 mM $H_2SO_4$ at 0.6 mL $min^{-1}$ with isocratic elution applied for 18 min[82]. A calibration curve of sodium pyruvate was prepared from 0.2 to 12 mM in each set of measurements. Similarly, the production and enantiomeric distribution of FAla was assessed by HPLC after derivatizing the samples. The analysis was performed on a Dionex UltiMate$^{TM}$ 3000 HPLC system equipped with a Dionex UltiMate$^{TM}$ 3000 Diode Array Detector; absorbance was continuously monitored at 310 nm. The quantification of FAla was based on a calibration curve of the corresponding pure enantiomers in a range of concentrations from 0.08 to 1.25 mM. Samples were loaded onto a Supelco$^{TM}$ Discovery$^{TM}$ HS F5-3 (15 cm × 2.1 mm, 3-µm) column (Sigma-Aldrich Co.) kept at 30 °C. The mobile phase was composed of 10 mM ammonium formate (pH = 3) and acetonitrile; an isocratic elution with 5% (v/v) acetonitrile was applied for the first 0.5 min, then the concentration of acetonitrile was increased to 60% (v/v) over a 6.5 min-gradient. The mobile phase was kept at 60% (v/v) acetonitrile for 2.5 min and then the system was re-equilibrated to the initial conditions applying an isocratic gradient of 5% (v/v) acetonitrile for 2.5 min. The flow was set to 0.7 mL $min^{-1}$ for the entire run. Data processing was carried out using the Chromeleon$^{TM}$ Chromatography Data System (CDS) software 7.2.9 (Thermo Fisher Scientific Co.).

### Mass spectrometry analysis

LC-MS(MS) analysis was performed using an UltiMate$^{TM}$ 3000 UHPLC binary system coupled to an Orbitrap Fusion$^{TM}$ mass spectrometer (Thermo Fisher Scientific Co.). Compound separation was achieved using a Waters$^{TM}$ Acquity UPLC BEH Amide (10 cm × 2.1 mm, 1.7-µm) column equipped with an Acquity UPLC BEH amide guard column kept at 40 °C. The mobile phases consisted of MilliQ$^{TM}$ water and 0.1% (v/v) formic acid (buffer A), and acetonitrile and 0.1% (v/v) formic acid (buffer B) at a flow rate of 0.35 mL $min^{-1}$. The elution was done through an initial step composed by 85% buffer B, held for 0.8 min, followed by a linear gradient to 50% buffer B over 3.2 min and held for 1 min and then linearly increased for 1 min to 30% buffer B before going back to initial conditions (the re-equilibration time was 3 min). The injection volume was set at 1 µL. The MS(MS) measurements were done in positive-heated electrospray ionization (HESI) mode with a voltage of 3500 V acquiring in full MS/MS spectra (data dependent

acquisition-driven MS/MS) with a m/z range of 50–500. The MS1 resolution was set at 120,000 and the MS2 resolution was set at 30,000. Precursor ions were fragmented by stepped high-energy collision dissociation (HCD) using collision energies of 20, 40 and 55 eV. The automatic gain control (AGC) target value was set at $4 \times 10^5$ for the full MS and $5 \times 10^4$ for the MS/MS spectral acquisition. Data analysis was performed using the FreeStyle 1.8 software (Thermo Fisher Scientific Co.).

## NMR analysis

$^{19}$F-NMR spectra were measured with a Bruker Avance III™ HD spectrometer (Bruker Corp, Billerica, MA, USA) operating at a magnetic field $B = 18.8\,T$ ($\nu_{19F} = 752.83\,MHz$) and equipped with a 5-mm TCI $^2$H/$^{19}$F-$^{13}$C-$^{15}$N CryoProbe[83]. All samples were measured at 30 °C unless otherwise noted. Each sample (400 μL) was added to a 5-mm NMR tube, shaken and quickly transferred to the spectrometer. For each spectrum in the time series, 100 free induction decay signals were recorded using a π/6 excitation pulse and 3 s of interscan delay, yielding a data point for every 5 min; 20-Hz sample rotation was applied during the measurements to increase mixing. Data analysis (including baseline correction and peak integration) was performed using the TopSpin 4.1.3 NMR software (Bruker Corp.). Due to the overlap between the multiplet signals from the FAla and FPyr, the total FAla signal intensity was determined by peak simulation of the most deshielded peak (calibrated to –228.760 ppm) of the *ddd* multiplet and multiplying this value by 8. The resulting FAla signal intensity was then subtracted from the full integral covering both FAla and FPyr multiplets to obtain the FPyr signal intensity. For the F$_3$Ala samples, simple peak integration was applied (using equal-sized integral regions) to extract the F$_3$Ala and F$_3$Pyr signal intensities[84].

## Data and statistical analysis

All the experiments reported were independently repeated at least three times (as indicated in the corresponding figure or table legend), and the mean value of the corresponding parameter ± standard deviation is presented unless indicated otherwise. Data analysis was performed with Prism 8 (GraphPad Software Inc., San Diego, CA, USA) unless differently specified.

## Reporting summary

Further information on research design is available in the Nature Portfolio Reporting Summary linked to this article.

## Data availability

All data generated or analyzed during this study are included in this published article (and its supplementary files). Additional data that support the findings of this study are available in the Supplementary Information. Materials are available from the corresponding author upon reasonable request. NMR spectra are within Supplementary Data 1, numerical source data are within Supplementary Data 2.

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

## Acknowledgements

M.N.D acknowledges the support received from the European Union's *Horizon2020* Research and innovation programme under the Marie Sklodowska-Curie grant agreement No. 713683 (*COFUNDfellowsDTU*) and from the VELUX Foundation under the Villum Experiment program (project No. 40979). The financial support from The Novo Nordisk Foundation (NNF10CC1016517 and NNF18CC0033664) and from the European Union's *Horizon2020* Research and Innovation Program under grant agreement No. 814418 (*SinFonia*) to P.I.N. is gratefully acknowledged. The NMR Center at DTU and the Villum Foundation are acknowledged for access to the 600 and 800 MHz NMR spectrometers. The responsibility of this article lies with the authors; the NNF and the European Union are not responsible for any use that may be made of the information contained herein.

## Author contributions

M.N.D.: Conceptualization, Investigation, Data curation, Methodology, Validation, Visualization, Writing – original draft; A.S.S.: Investigation, Data curation, Methodology; K.E.R.: Investigation, Data curation, Methodology; C.H.G.: Investigation, Data curation, Methodology; D.R.: Data curation, Methodology; P.I.N.: Conceptualization, Resources, Data curation, Funding acquisition, Supervision, Project administration, Writing – review & editing.

## Competing interests

The authors declare no competing interests.

## Ethics approval

The work presented in this article follows all prevailing local, national and international regulations and conventions, and normal scientific ethical practices.
