## [Peer Review File · Communications Chemistry]

Reviewers' comments:

Reviewer #1 (Remarks to the Author):

This is a very interesting project reported by Nieto-Dominguez et al. Fluorine is a magic element in organic chemistry. However, many fluorination syntheses suffer inefficiency. This report describes a one-pot biotransformation of enabling efficient production of stereospecific fluorinated alanine synthesis starting from fluorinated pyruvate. All of enzymes used in this report are well-characterized. However, combining these enzymes for stereospecific synthesis of fluorinated Ala is new. The manuscript was well-written and most of the experiments were well executed. This reviewer would like to think that this manuscript holds potential to be published in Comm Chem.

Comments:

1. the title is somehow a bit misleading. the synthesis of mono-FAla is in an excellent yield. however, F3Ala is not. I would suggest to change the title to reflect what was achieved.
2. the observation of the defluorination by racemase is interesting. however, it somehow distracts the main focus of the work as reflected in the title. it could be separated from this manuscript as it deserved its own right for a separated paper.
3. Can the authors try their best to identify the side products of the biotrans using F3PYr? F NMR indicated that one of the side products was even higher than the target. what are they? Are they still trifluorinated compounds? considering the simple reaction conditions of biotrans, the possibility of side products could be limited.
4. The authors need to pay attention to Figures S4 and S5. It is difficult to judge what is going on in these two figures.

Reviewer #2 (Remarks to the Author):

Dear Authors

The biosynthesis of fluorinated compounds in a selective manner is one of major challenges in biotechnology. There are a few representative works published, but mostly limited to a few substrates or positions to be tackled, respectively, it is of importance to improve and widen the portfolio of those reactions. The provision of non-canonical amino acids as here fluorinated alanine provides one possible avenue to add to the field.

Overall, I must say the manuscript is well written and for the expert as well as for the general reader easy to follow. It expands the set of available fluorinated compounds to be used in biotechnology by a well-selected enzymatic approach.

I have some minor comments which I hope are supportive to improve the manuscript:

- In the results section it might be more important to add information on turnover frequencies and total turnover numbers; that will help to later have comparable work in those directions.
- The control of enantiomeric excess; it looks as you have a major peak and a small shoulder running in

front; can you elaborate on that?

- The term “protein expression” should be changed to “gene expression” as only genes can be expressed and as a result a protein can be produced.
- Please provide a table or overview of the molar extinction coefficients of the proteins including the tags etc. which had been used to calculate the protein content of samples.
- The methods are clear, I just recommend to not only to use the molar extinctions for protein conc. determination as this might be effected by the buffers and impurities taken along the protein purification.

Supplemental material

- This is important to the manuscript and therefore should be published along without any limitations.
- Fig. S4 and S5; in the PDF file provided the MS-spectra are not visible.

Response to the Reviewers' comments and how they have been addressed in the revised version of COMMSCHEM-23-0589 (in blue)

Reviewer # 1

This is a very interesting project reported by Nieto-Dominguez *et al.* Fluorine is a magic element in organic chemistry. However, many fluorination syntheses suffer inefficiency. This report describes a one-pot biotransformation of enabling efficient production of stereospecific fluorinated alanine synthesis starting from fluorinated pyruvate. All of enzymes used in this report are well-characterized. However, combining these enzymes for stereospecific synthesis of fluorinated Ala is new. The manuscript was well-written and most of the experiments were well executed. This reviewer would like to think that this manuscript holds potential to be published in *Comm. Chem.*

[1] The title is somehow a bit misleading. The synthesis of mono-FAla is in an excellent yield. However, F3Ala is not. I would suggest to change the title to reflect what was achieved.

We apologize for the lack of accuracy. The title has been modified to **Enzymatic synthesis of mono- and trifluorinated alanine enantiomers expands the scope of fluorine biocatalysis**, which reflects more faithfully the main results of our study.

[2] The observation of the defluorination by racemase is interesting. However, it somehow distracts the main focus of the work as reflected in the title. It could be separated from this manuscript as it deserved its own right for a separated paper.

We thank the reviewer for the suggestion. As the reviewer rightfully mentioned, the main goal of the manuscript is to study the enzymatic production of FAla. However, the (unexpected) defluorination capacity of the racemase from *Streptomyces lavendulae* was observed in connection with our efforts to produce (S)-FAla in a two-enzymatic step using an alanine dehydrogenase. Since the ability of certain ALDHs to catalyze the reductive amination of FPyr is well known, coupling this activity with novel alanine racemase emerges as a straightforward approach to support the synthesis of fluorinated D-alanine. Therefore, regardless the interest of ^SAlaR as a defluorination enzyme, we think that the results reported in this manuscript will be valuable for future research attempting to produce fluorinated D-amino acids through the stereochemical inversion of their better-known L-counterparts. Consequently, we prefer to maintain these results in the present manuscript.

[3] Can the authors try their best to identify the side products of the biotrans using F₃Pyr? F-NMR indicated that one of the side products was even higher than the target. What are they? Are they still trifluorinated compounds? Considering the simple reaction conditions of biotrans, the possibility of side products could be limited.

We thank the reviewer for highlighting this interesting result. Both the chemical shifts and the fact that the detected signals are singlets suggest the presence of a -CF₃ group. In addition, the spectra do not show the presence of free fluoride anion, and it is expected for the -CF₃ group to be present in all side products. The current data does not allow for quantification for side products, but we note that they are minor (the signal is not proportional to concentration in these measurements). The corresponding discussion of these results can be found on Page 9 lines 1-5 in the main text, and additional spectra have been added to the Supplementary Material (Fig. S6).

[4] The authors need to pay attention to Figures S4 and S5. It is difficult to judge what is going on in these two figures.

We apologize for the lack of clarity. The quality of the figures has been improved to make them easier to understand.

Reviewer # 2

The biosynthesis of fluorinated compounds in a selective manner is one of major challenges in biotechnology. There are a few representative works published, but mostly limited to a few substrates or positions to be tackled, respectively, it is of importance to improve and widen the portfolio of those reactions. The provision of non-canonical amino acids as here fluorinated alanine provides one possible avenue to add to the field. Overall, I must say the manuscript is well written and for the expert as well as for the general reader easy to follow. It expands the set of available fluorinated compounds to be used in biotechnology by a well-selected enzymatic approach. I have some minor comments which I hope will help to improve the manuscript:

[1] In the results section it might be more important to add information on turnover frequencies and total turnover numbers; that will help to later have comparable work in those directions.

We thank the Reviewer for the suggestion. Both the turnover frequency (TOF) and the total turnover number (TTN) are informative parameters¹ and, following the reviewer's suggestion, we have modified the manuscript to add the calculation of TOF and TTN for the production of FAla using either ^{Vp}ALDH or ^{Sd}DAPDH coupled with the regeneration of the NAD(P)H cofactor (Page 13, lines 8-11). Since the final yields were similar for both 12 h and 24 h, the shortest reaction time was selected to maximize the TOF value.

[2] The control of enantiomeric excess; it looks as you have a major peak and a small shoulder running in front; can you elaborate on that?

The small peak is present in both the (*R*)-3-fluoroalanine and (*S*)-3-fluoroalanine commercial standards when analyzed individually. Such peak was also detected for the corresponding control in which the derivatization procedure was carried out using MilliQ water instead of any enzymatic sample. These observations suggest that the peak may be related to the derivatization reagent itself. The presence of additional peaks when using the Marfey's protocol has been reported previously, and it has been attributed to the excess of reagent employed for the derivatization procedure and to the formation of side-products due to its hydrolysis^{2,3}. We have modified the main text of the manuscript to include a summary of this discussion (Page 12, lines 24-28) and a new supplementary figure (Fig. S12) has been added with the relevant controls to highlight the peak mentioned by the reviewer at 5.6 min and another small peak at 5.3 min which can be accounted for under the same framework.

[3] The term "protein expression" should be changed to "gene expression" as only genes can be expressed and as a result a protein can be produced.

We thank the reviewer for noticing this error. The manuscript has been corrected as suggested.

[4] Please provide a table or overview of the molar extinction coefficients of the proteins including the tags etc. which had been used to calculate the protein content of samples.

Following the reviewer's suggestion, we have included a new supplementary table (Table S3) showing the full amino acid sequence of the proteins in this study, together with their predicted molar extinction coefficients. This additional information is also referred to in the main text (Page 17, lines 3-5).

[5] The methods are clear, I just recommend to not only to use the molar extinctions for protein conc. determination as this might be effected by the buffers and impurities taken along the protein purification.

We thank the reviewer for the suggestion. Measuring the absorbance at 280 nm (A_{280}) and calculating the concentration based on the corresponding molar extinction coefficients (ϵ_{280}) is a standard method for protein quantification. We agree that the presence of impurities can potentially interfere with the measurements, and it is not a method suitable for complex samples such as cell lysates. However, the protocol of this study includes the buffer exchange of the purified proteins using centrifugal filters of 10

kDa-pore size. The new buffer composition does not contain any reagent expected to interfere with the A280 measurements and no significant differences were observed when equivalent protein preparations in water were analyzed. Moreover, the SDS-PAGE analysis of the isolated proteins shows a high level of purity. Consequently, we consider that our approach for protein quantification was adequate for the purpose of this study.

Supplemental material

- This is important to the manuscript and therefore should be published along without any limitations.
- Fig. S4 and S5; in the PDF file provided the MS-spectra are not visible.

We apologize for the error. The quality of the figures has been improved to ensure the spectra is still visible after the conversion into a PDF file.

REFERENCES

1. Kozuch, S. & Martin, J.M.L. "Turning over" definitions in catalytic cycles. *ACS Catal.* **2**, 2787-2794, (2012).
2. B'Hymer, C., Montes-Bayon, M. & Caruso, J.A. Marfey's reagent: Past, present, and future uses of 1-fluoro-2,4-dinitrophenyl-5-L-alanine amide. *J. Sep. Sci.* **26**, 7-19, (2003).
3. Bhushan, R. & Brückner, H. Use of Marfey's reagent and analogs for chiral amino acid analysis: Assessment and applications to natural products and biological systems. *J. Chromatogr. B Analyt. Technol. Biomed. Life Sci.* **879**, 3148-3161, (2011).

Reviewer #2 (Remarks to the Author):

Dear Authors

The revised manuscript reads well, all suggestions and questions from the previous review process have been used to substantially improve the manuscript. From my point of view the

- methods are clear and scientific sound
- work is reproducible
- conclusions are justified by numerous results

And thus the work contributes with novel findings to field!

Sincerely!